# Non-Invasive Biomarkers for Immunotherapy in Patients with Hepatocellular Carcinoma: Current Knowledge and Future Perspectives

**DOI:** 10.3390/cancers14194631

**Published:** 2022-09-23

**Authors:** Maria Pallozzi, Natalia Di Tommaso, Valeria Maccauro, Francesco Santopaolo, Antonio Gasbarrini, Francesca Romana Ponziani, Maurizio Pompili

**Affiliations:** 1Internal Medicine and Gastroenterology-Hepatology Unit, Fondazione Policlinico Universitario Agostino Gemelli IRCCS, 00168 Rome, Italy; 2Translational Medicine and Surgery Department, Università Cattolica del Sacro Cuore, 00168 Rome, Italy

**Keywords:** biomarkers, gut microbiota, hepatocellular carcinoma (HCC), immunotherapy, liquid biopsy, PD-1, PD-L1

## Abstract

**Simple Summary:**

The search for non-invasive biomarkers is a hot topic in modern oncology, since a tissue biopsy has significant limitations in terms of cost and invasiveness. The treatment perspectives have been significantly improved after the approval of immunotherapy for patients with hepatocellular carcinoma; therefore, the quick identification of responders is crucial to define the best therapeutic strategy. In this review, the current knowledge on the available non-invasive biomarkers of the response to immunotherapy is described.

**Abstract:**

The treatment perspectives of advanced hepatocellular carcinoma (HCC) have deeply changed after the introduction of immunotherapy. The results in responders show improved survival compared with Sorafenib, but only one-third of patients achieve a significant benefit from treatment. As the tumor microenvironment exerts a central role in shaping the response to immunotherapy, the future goal of HCC treatment should be to identify a proxy of the hepatic tissue condition that is easy to use in clinical practice. Therefore, the search for biomarkers that are accurate in predicting prognosis will be the hot topic in the therapeutic management of HCC in the near future. Understanding the mechanisms of resistance to immunotherapy may expand the patient population that will benefit from it, and help researchers to find new combination regimens to improve patients’ outcomes. In this review, we describe the current knowledge on the prognostic non-invasive biomarkers related to treatment with immune checkpoint inhibitors, focusing on serological markers and gut microbiota.

## 1. Introduction

In recent years, the management of hepatocellular carcinoma (HCC) has dramatically changed. The recent update of the current guidelines emphasizes that the expansion of the drug armamentarium of tyrosine kinase inhibitors [1,2,3] (TKIs) and the introduction of immunotherapy as a first-line treatment has improved survival results, even in advanced stages [4]. Several trials have tested the efficacy of immune checkpoint inhibitors (ICIs), a class of drugs blocking the programmed cell death receptor 1 (PD-1), programmed cell death ligand 1 (PD-L1), cytotoxic T-lymphocyte antigen 4 (CTLA-4), lymphocyte activation gene 3 (LAG-3), and mucin domain molecule 3 (TIM-3) in monotherapy or in combination with other ICIs or TKIs [5]. The results of these studies have demonstrated the superiority of ICI-based therapies versus Sorafenib in terms of overall survival (OS) and progression free survival (PFS) [6,7]. The combination of Atezolizumab plus Bevacizumab or Durvalumab [8] plus Tremelimumab has been recently recommended as a first-line regimen in advanced HCC [4]. Other treatment regimens such as Nivolumab plus Ipilimumab [9] or Pembrolizumab [10] received Food and Drug Administration (FDA)-accelerated approval as second line therapies in patients previously treated with Sorafenib, even if the best sequential strategy for the administration of these drugs has yet to be well defined, and even though Pembrolizumab did not satisfy the pre-planned statistical threshold of effectiveness in spite of the improved OS and PFS vs. a placebo in a phase III trial involving patients progressing after Sorafenib [11]. Other ongoing studies evaluate the role of ICIs in the neoadjuvant or adjuvant setting in combination with locoregional treatments or surgical resection [12,13,14].

Despite the significant results obtained with immunotherapy, not all patients receive adequate benefits from this type of treatment [15,16]. Multiple factors may be associated with this variable effect, depending on the characteristics of the host and tumor microenvironment (TME), the latter being only marginally involved in the treatment allocation for HCC. This is the result of the biopsy-sparing diagnostic approach to this type of tumor, which has severely limited the progress in prognostic stratification in recent years, generating a gap between the clinical characteristics of patients and the histological counterpart of HCC [16]. Hence, it is crucial to identify non-invasive biomarkers to prognosticate the probability of response to ICIs, as this would allow patients to be directed to the most appropriate drug or combination of drugs. In light of these considerations, this review will address the current knowledge and future perspectives on the non-invasive biomarkers of a response to immunotherapy in patients with HCC.

## 2. Clinical Usefulness of Non-Invasive Biomarkers

Tumor biomarkers are cellular and molecular products linked directly or indirectly to the presence of cancer cells that are an expression of the tumor’s intrinsic characteristics and can be identified, measured, and analyzed by specific tests [17,18]. They can be used for multiple purposes, mainly for the early detection of a tumor, but also for defining its biological behavior and aggressiveness, and for assessing the response following a therapeutic intervention [19]. In the past few years, HCC biomarkers have been prevalently derived from histological analysis, but currently there is an increasing interest in developing novel techniques to extract information on tumor biology directly from patient’s body fluids in a non-invasive way [20,21,22]. Indeed, liver biopsy is invasive and requires the use of structural resources with consequent costs; furthermore, it renders a timely picture of the TME or genomic landscape and cannot capture dynamic changes over time. A liquid biopsy can overcome these limitations. It is a non-invasive approach of detecting tumor-derived products that can be performed at different time points, drawing a personalized and dynamic picture of HCC’s evolution and response to therapy [23]. There are various tumor by-products that can be measured in the bloodstream alone or in combination [24], some of which are well-known and traditionally linked to the mechanisms of carcinogenesis (e.g., cytokines and alpha-fetoprotein), while others have only recently been studied (e.g., circulating tumor cells, DNA, RNA, and exosomes).

## 3. Traditional Non-Invasive Biomarkers of Response to ICIs

### 3.1. Cytokines, Immune Checkpoints, and Immune Cells

Circulating soluble factors, such as cytokines, have been evaluated as biomarkers over the course of ICI therapy [25]. An evaluation of the baseline levels of 34 circulating serum cytokines and chemokines of patients receiving Atezolizumab plus Bevacizumab highlighted that only interleukin 6 (IL-6) and interferon alpha (IFN-alpha) were related to disease progression [26]. IL-6 is an inflammatory cytokine involved in liver regeneration, but also shows an oncogenic role [27,28], as it is involved in the recruitment of myeloid-derived suppressor cells, thereby favoring tumor immune escape [29]. It can be simply detected by an enzyme-linked immunosorbent assay (ELISA) [25,26].

A high serum level of IL-6 was also associated with worse PFS and OS and was more frequently found in females; in patients with high Aspartate Aminotransferase (AST), Alpha Fetoprotein (AFP), and Des-gamma-carboxy prothrombin (DCP) values; and those with reduced liver function. A limitation of this study is the small number of Asian patients enrolled. Moreover, IL-6 serum levels are influenced by inflammatory conditions, reducing their specificity [25,26,27,28,29].

In another study, performed to evaluate the correlation between circulating biomarkers and the response to the anti-PD-1 Pembrolizumab after 60–90 days of treatment in patients with advanced HCC, among several plasma biomarkers, only transforming growth factor (TGF)-beta serum levels significantly differed between the responders and non-responders (141.9 pg/mL vs. 1071.8 pg/mL) [30]. Plasma levels of TGF-beta higher than 200 pg/mL indicated a poor treatment response and a reduced PFS and OS (*p* = 0.003). Circulating biomarkers, assessed by ELISA, demonstrated that higher levels of PD-1 and PD-L1 were associated with the upregulation of IFN-gamma and IL-10 (*p* < 0.05); patients with a high tumor expression of PD-1 showed higher serum levels of IFN-gamma and IL-10, but plasma PD-L1 did not correlate with tumor PD-L1 expression. Changes in plasma biomarkers were also observed after treatment: only Chemokine (C-X-C motif) ligand 9 (CXCL9) increased regardless of the response (*p* = 0.008) [30].

The impact of PD-L1 expression in HCC TME and its correlation with prognosis has not been clarified yet. PD-L1 expression is not homogeneous in tumor tissue, varying over time depending on several stimuli; the different assays used to determine PD-L1 expression in different studies may also be partially responsible for these results [25,31,32].

PD-L1 expression on peripheral immune cells has been previously reported in patients with HCC [33]. Recent studies demonstrated that the response to ICIs may be influenced by the prevalence of PD-L1 expression on cluster differentiation (CD) 4 T lymphocytes before treatment [34]. Furthermore, immunotherapy may influence the expression of PD-1 and PD-L1 during systemic treatment on double positive CD4/CD8 T cells and on CD4 T cells in responder patients compared to non-responders, even in the absence of differences at the baseline [35,36]. Soluble PD-L1 (sPD-L1) has been tested in patients with renal cell carcinoma or melanoma prior to, and at two time points during, treatment with Nivolumab; a progressive or stable disease was associated with an increase in sPD-L1 [35]. The reliability of these results is affected by the absence of standardized reference levels for sPD-L1 and of pre-established cut-off levels for prognosis and response prediction. Moreover, ELISA results may vary depending on the assay kit [35]. In HCC patients, a meta-analysis demonstrated that a high sPD-L1 level correlates with a shorter survival (HR: 2.93; 95% CI: 2.20–3.91; *p* < 0.00001) [36]. In a study conducted in early HCC patients who underwent surgical liver resection, the persistence after treatment of sPD-L1 indicated a poor outcome, suggesting the possibility to apply sPD-L1 analysis for identifying patients eligible for adjuvant immunotherapy [37]. Overall, the predictive value of sPD-L1 during ICIs therapy remains controversial and further studies are needed [38].

Peripheral Blood Mononuclear Cells (PBMC) reflect the changing landscape of tumor-associated immune cells before and after immunotherapy. They can be easily identified and characterized using flow cytometry, with a subset of antibodies binding to different CD proteins [39]. In patients with HCC, the response to ICIs is influenced by the number of peripheral effector T cells, which are more prominent in the presence of CTLA-4 and the inducible Co-stimulator ICOS on PBMC surfaces, regardless of the etiology of liver disease. Moreover, anti-PD-1 agents upregulate cytotoxic T cell effectors with a memory phenotype, also reducing PD-1 expression on their surface, and cause the downregulation of B cells [40]. The presence of CD14+ CD16− Human Leukocyte Antigen-DR isotype (HLA-DRhi) monocytes in peripheral blood before ICIs treatment is another marker of a favorable outcome; monocytes presenting these receptors can favor the infiltration of T cells in tumor tissue, leading to the activation of T cell effectors against cancer [41]. Other studies demonstrated that baseline CD4+ PD-1+ cells predict a successful response to the anti-CTLA-4 Tremelimumab, and that a lower percentage of T regulatory cells as well as higher levels of double positive CD4/CD8 T cells are associated with the response to anti-PD-L1 inhibitors [40,42]. PBMCs obtained before and after 6 weeks of therapy with Pembrolizumab in patients with advanced HCC showed that the immune cell distribution was not homogeneous, with levels of CD8 T cells and CD4 T naive cells that differed among the samples, suggesting a role for this imbalance in response modulation [43]. After therapy, an increased number of activated CD4 and CD8 T cells and a reduction in CD4 and CD8 naive T cells was observed. Moreover, the responders had higher levels of cytotoxic CD8 T cells, while the poor responder patients expressed molecules associated with neutrophil-associated pathways. In another study, Nivolumab therapy was associated with an elevation of CD8 αβT cells after 4 weeks of therapy; these cells showed lower levels of PD-1 expression compared to that belonging to patients with disease progression. No significant alteration in regulatory T cells was observed, and the patients achieving disease control (DC) maintained a persistent expression of PD-L1 on their monocytes after 28 and 42 days of treatment. The post treatment changes in the PD-L1 positivity of the patients’ monocytes differed among responders and non-responders after 28 and 42 days. Finally, the low pretreatment expression of PD-1 on peripheral B cells was associated with DC [42]. In a study by Macek Jilkova et al., the composition and expression of CD in blood lymphocytes, natural killer (NK) T cells, and NK cells were evaluated and compared to T cells and NK cells within the liver in 21 patients with advanced HCC treated with Sorafenib [44]; the CD31+ and CD56+ T-cells found in the blood accounted for more than 60% of all CD45-high lymphocytes, whereas in the liver, their frequency decreased to 50%. Conversely, CD3+ CD56+ cells (NKT and CD3brightCD56+ T cells) and CD3- CD56+ NK cells represented a significantly smaller population in the blood (7% and 11%), but their frequency increased up to 21% and 21% in nontumor liver tissues and 19% and 17% in tumor liver tissues, respectively. Dividing the population according to their progression after 16 weeks, the elevation of IL-10 in the blood samples correlated with a higher CD4/CD8 ratio in tumor and non-tumor tissues and a poor prognosis. The expression of the immune checkpoint molecules LAG3 and TIM3 on circulating T cells was upregulated in non-responders and associated with poor survival [44].

### 3.2. Neutrophil to Lymphocyte Ratio, Platelet to Lymphocyte Ratio, Prognostic Nutritional Index, and Their Combined Prognostic Value

The Neutrophil to Lymphocytes Ratio (NLR) has been used to evaluate the risk of mortality in patients with liver disease [45,46], and has also been tested as a prognostic biomarker for immunotherapy in several studies. A decline in the NLR in patients with HCC receiving anti PD-1 therapy was associated with a better response to treatment and improved survival [46,47]. In patients receiving Atezolizumab plus Bevacizumab, the baseline NLR (cut-off 3.21) resulted to be an independent predictor of response, and was directly linked with PFS [48]; in another study, a low NLR was associated with a better OS [49]. Similarly, a high Platelet to Lymphocytes ratio (PLR), an inflammatory marker already correlated with prognosis in patients with HCC [46,50], has been associated with poor prognosis [46]. Finally, a high prognostic Nutritional Index (PNI), obtained by multiplying the serum albumin (g/dL) by the total lymphocyte count, was associated with a response to anti-PD-1 therapy and better survival [51].

Based on these results, a study combined NLR, PLR, and PNI to evaluate their roles as predictors of response to immunotherapy [52]. The blood samples of 362 HCC patients were collected; 74.3% of themhad advanced disease, and the most frequent liver disease etiology was the hepatitis C virus. Eighty percent of the patients received anti-PD-1 monotherapy, while the others received a combination of anti-PD-1 plus anti-CTLA-4 or TKIs. The median NLR value in the whole cohort was 3.55 (0.06–25.3), the median PLR value was 137.32 (0.17–1100), and the median PNI value was 40.29 (1.11–1270). The previously indicated thresholds for a high mortality risk (NLR ≥ 5, PLR ≥ 300, and PNI < 45) were associated with the presence of portal vein thrombosis, a worse performance status, a more advanced Barcelona Clinic Liver Cancer (BCLC) stage, and worse OS and PFS (limited to PLR and PNI). In the multivariate analysis, an NLR ≥ 5 and a PLR ≥ 300 remained independent prognostic factors for OS (HR 1.73, 95% CI 1.23–2.42, *p* = 0.002 and HR 1.60, 95% CI 1.6–2.40, *p* = 0.020, respectively), while the PLR was the only independent predictor of PFS. These results were partially confirmed by a Chinese study reporting that all the inflammation-based prognostic scores showed good discriminatory ability in OS, but only the PNI score was an independent predictor for OS in multivariate analysis [53].

Finally, albumin, lactate dehydrogenase (LDH), and the NLR were combined in a score named the Gustave Roussy immune score (GRIm-Score), which was able to stratify patients receiving ICIs into responders and non-responders with respect to several tumors [54]. To better mirror the characteristics of HCC, the GRIm-Score was implemented with other circulating markers that resulted independent prognostic factors of survival in a multivariate analysis; therefore, the HCC-GRIm-Score was obtained, including albumin (<35 g/L = 1), LDH (>245 U/L = 1), NLR (≥4.8 = 1), Aspartate Aminotransferase to Alanine Aminotransferase AST-to-ALT ratio (≥1.44 = 1), and total bilirubin (≥22.6 umol/L = 1). Lower scores (from 0 to 2 points) resulted in a better OS [55]. Despite the remarkable results highlighted by these studies on the utility of a combinatory approach of non-invasive biomarkers to improve sensitivity, further and larger clinical trials are needed to apply them in clinical practice.

### 3.3. Alpha-Fetoprotein

Serum alpha-fetoprotein is the HCC-related biomarker that is most used in clinical practice [56,57]. Studies showed that early AFP reduction is associated with a good prognosis in patients with HCC receiving ICIs [58,59,60,61]. Previous in vitro studies demonstrated a pro-oncogenic role of AFP in inducing resistance to tumor necrosis factor (TNF) cytotoxic activity, promoting protein kinase A activity and the expression of the pro-oncogenic proteins p53 and p21 [62,63]. In addition, NK cells’ activity has been shown to be impaired in the presence of AFP-treated dendritic cells (DCs), which decreases IL-12 production [64]. AFP also exerts a pro-oncogenic priming in both animal and human models of HCC by reducing the FAS/FAS-associated death domain protein (FADD) apoptotic pathway through the modulation of Human Antigen R RNA-binding protein, resulting in the promotion of tumor growth [65]. For these reasons, targeting AFP could be a new therapeutic strategy in the treatment of HCC, and an ongoing trial is evaluating the efficacy of T cells’ reprogramming against AFP in AFP-expressing HCC patients [66].

AFP has the advantage of being detectable via traditional methods, such as radio- and fluorescent-immunoassays, which are available in almost all laboratories [67]. However, current guidelines highlight the suboptimal performance of AFP for HCC screening due to several limitations [68,69]. In particular, when used as a diagnostic test, AFP shows good sensitivity but low specificity with a cut-off of 20 ng/mL, while the sensitivity decreases by increasing the cut-off to 200 ng/mL, although improving specificity [68]. Moreover, many HCCs do not express AFP; on the contrary, liver cirrhosis and other intrahepatic and extrahepatic non-HCC tumors may be associated with an increase in AFP serum levels [70]. Finally, when combined with ultrasonography, AFP increases the ability to detect previously unidentified nodules by 6–8% [68].

Models combining AFP with other biochemical parameters have reported promising results with respect to predicting ICIs response. One of them that includes AFP and C-reactive protein (CRP), named the CRAFITY score, has been recently evaluated in HCC patients receiving anti-PD-1 immunotherapy. The patients with a CRAFITY score of 0 showed a better radiological response and OS compared to patients with a CRAFITY score ≥ 1 [71,72].

Another study combined the CRAFITY score with AFP decline after 6 weeks of treatment with Atezolizumab plus Bevacizumab, generating a classification named CAR (CRAFITY score and AFP Response) [73]. The patients were divided into three classes: those with a low CRAFITY score and good AFP response (class I), either a high CRAFITY score or an unsatisfactory AFP response (class II), and a high CRAFITY score and an unsatisfactory AFP response (class III). The Median Objective Response Rate (ORR) was better in class I than in classes II or III (35% vs. 18.2% vs. 0%). The patients in class I had the best OS and PFS, followed by those in classes II and III (the median OS of class I did not reach the median time exceeding the 12 months of follow-up vs. 11.1 months vs. 4.3 months, *p* < 0.001; the median PFS was 7.9 months vs. 6.6 months vs. 2.6 months, *p* = 0.001).

Another study analyzed the prognostic ability of AFP plus prothrombin induced by vitamin K absence-II (PIVKA-II) in HCC patients treated with anti-PD-1 therapy [74]. Reductions in AFP and PIVKA-II of more than 50% from the baseline levels at 6 weeks of treatment were associated with a favorable ORR and a better OS and PFS. The combination of these results with the Albumin Bilirubin score (ALBI) was included in the AAP score; patients with an AAP score ≥ 2 points showed a significantly longer PFS and OS. Lower serum levels of AFP and soluble intercellular adhesion molecule 1 (sICAM-1)—a soluble factor derived from endothelial cells and involved in inflammatory responses, which is absent in normal hepatocytes [75,76]—were associated with a response to immunotherapy in a recent analysis of the Cancer Genome Atlas Liver Hepatocellular Carcinoma. AFP and ICAM-1 upregulation correlate with an immunosuppressive TME in which CD4 T cells, macrophages, and monocytes are predominant, and the expression of immune checkpoints such as T cell immunoglobulin and ITIM domain (TIGIT), Hepatitis A virus cellular receptor 2 (HAVCR2), PD-1, CTLA4, and LAG3 is enhanced [77].

In conclusion, AFP is a definitively cheap non-invasive biomarker, but its combination with other serum markers [71,72,73,74] deserves further investigation to improve diagnostic accuracy.

A summary of the main findings of the available studies on traditional non-invasive biomarkers described herein is reported in Table 1.

## 4. Novel Biomarkers

### 4.1. Circulating Tumor DNA

Circulanting Tumor DNA (ctDNA) are cell-free DNA products released by tumor cells in the bloodstream during apoptosis or necrosis that are rapidly cleared by macrophages [78]. ctDNA is detected using real-time PCR and digital PCR, which are able to recognize DNA aberrations in target genes in the presence of a specific probe; this limitation is overcome by Next Generation Sequencing (NGS), which allows for the identification of new genomic aberrations that may cause resistance to therapy [79]. Another method is the Sequenom’s (San Diego, CA, USA) MassARRAY Compact system, which is able to measure the direct mass of nucleic acids with high precision, in addition to quantifying gene expression and identifying genotypes and the methylation of ctDNAs [80]. The concentration of ctDNA in peripheral blood is higher in patients with HCC than in cirrhotic patients without a tumor and healthy controls. Interestingly, tumor size, extrahepatic spread, and vascular invasion are associated with higher ctDNA levels [81]. Tumor ctDNA fragments are longer when compared to circulating DNA derived from other apoptotic host cells; indeed, the ratio of ctDNA to the whole circulating DNA length, called DNA integrity, has been used as a marker for the early diagnosis of HCC [82]. 

A qualitative analysis of somatic gene mutations in HCC-derived ctDNA fragments showed that several oncogenes and tumor suppressor genes such as ARIDNA, tumor protein 53 (TP53), catenin (cadherin-associated protein), and beta 1 (CTNNB1) are involved, and the same mutations were found in tumor tissue in 63% of cases [82,83]. The presence of Telomerase reverse Transcriptase gene (TERT) mutations in ctDNAs are associated with vascular invasion [84]. Mutations in the PI3K/mTOR pathway correlate with a shorter PFS in patients treated with TKIs but not with ICIs [84], whereas mutations in MutL homolog 1 are linked to a worse OS [85]. 

The detection of a TP53 R249S mutation in ctDNA after HCC resection is a marker of a poor disease-free survival (DFS) [86]. ctDNA detectable after curative treatment has been related with microvascular invasion and may predict tumor recurrence and extrahepatic spread [87]. Accordingly, preoperative ctDNA detection was associated with larger HCCs, multiple lesions, microvascular invasion, and shorter PFS and OS [81]. A high tumor mutational burden (TMB) in the tumor genome was associated with an effective immune response against several tumors, due to the increased neoantigen load [88,89]. In addition to a TMB assessment on liver tissue, an emerging blood-based technique has been described and performed on ctDNA. Blood TMB (bTMB) accurately reflects tissue TMB, as described for several tumors. Recently, a commercial ctDNA platform for quantifying bTMB from blood has proven to be sensitive and reproducible, with an optimal ctDNA TMB cutoff of ≥20 mut/Mb that predicted positive results with Durvalumab plus Tremelimumab compared to chemotherapy in non-small cell lung cancer [90].

In a pilot study, the analysis of a cohort of patients with advanced solid tumors treated with immunotherapy demonstrated the correlation between bTMB and tTMB regardless of tumor histology [91]. However, the patients with higher levels of bTMB did not achieve a better OS, but the heterogeneity of the cohort may explain this result. Interestingly, an exploratory analysis from the same study demonstrated a reduction in the ctDNA mutant allele frequency (MAF) over the treatment period in the responders. Similarly, ctDNA MAF has been reported to change dynamically after surgery and to correlate with recurrence-free survival and OS [92]. Despite these promising results, at present, the application of ctDNA as a biomarker is limited, mainly due to the lack of standardized procedures for sample preparation and the difficulty of determining the assay when the ctDNA levels are very low [93].

### 4.2. Circulating Tumor Cells

Circulating Tumor Cells (CTCs) are rounded, nucleated cells released into the bloodstream from the primary tumor site or from metastatic sites [94]. CTCs express epithelial proteins such as Epithelial Cell Adhesion Molecules (EpCAM); cytokeratin 8, 18, or 19; and stem cells markers, while they lack the CD45 antigen. They can be found as single cells or in clusters, according to their mono- or oligo-clonal origin; CTC clusters are more prone to seeding, as they express transcription factors of genes that enhance proliferation. 

CTCs can be isolated using different techniques, categorized as follows: (1) physical methods, according to size (filtration-based devices), electric charge (electrophoresis), density (Ficoll centrifugation), migratory capacity, and deformability; (2) biological methods, based on antigen–antibody binding, for a qualitative analysis of the proteins expressed on CTCs surfaces. Specifically, biological methods use antibodies against specific tumor biomarkers, such as EpCAM, which is not expressed in blood cells but only in cells of an epithelial origin. The EpCAM-based CellSearch platform (Veridex LLC, Raritan, NJ, USA) has been approved by the FDA for CTC detection. This method has a limitation because although EpCAM is expressed in most epithelial-derived cells but not in blood cells, tumor cells who undergo endothelial-mesenchymal transition are not detected; furthermore, the absence of standardization limits the diffusion of CTC detection [95]. To overcome this issue, CTCs can be detected with the Canpatrol platform, a multiplex RNA in situ hybridization against EpCAM and several epithelial cytokeratines [96]. More recently, a microscale technology was developed to increase sensitivity and identify lower levels of CTCs [97]. CTCs are increased in patients with HCC, and their levels correlate with the prolongation of the disease, tumor stage, and AFP serum levels [98]. The number of CD3, CD4, and CD8 T cells negatively influences the presence of CTCs, while the number of T regulatory cells is associated with a greater number of CTCs [78]. 

Despite their low concentration in peripheral blood (about 5–50 CTCs in 7.5 mL of blood) [99] and their short half-life (2.5 h) [100], several studies of different tumors showed that CTCs may be useful as prognostic markers after treatment. Indeed, it has been reported that after HCC surgical resection or locoregional treatment, the CTCs concentration drops [101], whereas any increase after treatment is associated with a higher risk of tumor recurrence [102]. Studies in several cancers showed that CTCs expose immune checkpoints such as PD-L1, PD-L2, and CTLA-4 on their membrane, so they have been evaluated as biomarkers to identify patients suitable for immunotherapy [103]. In a phase I trial, the total number of CTCs and the PD-L1 expression on CTCs was evaluated in patients with advanced gastrointestinal cancers, including HCC, at baseline and following anti-PD-1 therapy [104]. The patients were divided into four categories based on their PD-L1 expression at baseline (negative, low, medium, and high); 74% of the patients showed PD-L1-high CTCs, which correlated with a good treatment response, while the persistence of PD-L1-high CTCs after therapy was associated with a poor outcome. Reductions in total CTCs and in PD-L1 expression on CTCs from the baseline levels were also reported in patients with a stable disease. Even though a positive correlation between baseline PD-L1-high CTCs and disease status was not detected, the average percentage of PD-L1-high CTCs among the total number of CTCs in the patients with disease control was significantly higher compared to the patients with a lower expression of PD-L1-high CTCs; moreover, the PD-L1-high CTC/total CTCs ratio was higher in the responders. Further, the absence of PD-L1-high CTCs at baseline was associated with a higher risk of progression during anti-PD1 therapy. Taken together, these results suggest that the presence and distribution of PD-L1-high CTCs could be a better biomarker in predicting PD-1 therapy response compared to PD-L1 positive CTCs. Another study including only HCC patients confirmed that PD-L1+ CTCs identified responders to ICIs [105].

Despite the promising role of CTCs as a biomarker for HCC, their low concentration in the bloodstream and short half-life can limit the reliability of their respective assay. Moreover, the validated platform has a weak performance with respect to the recognition of CTCs with markers of endothelial-mesenchymal transition. The lack of standardized clinical trials with large cohorts of patients is another concern that should be overcome [106].

### 4.3. Extracellular Vesicles

Extracellular vesicles (EVs) are nanoparticles that have heterogeneous functions, including the ability to modulate inter-cellular communication, and could influence several processes, such as inflammation and tumorigenesis [107,108]. EVs can be classified in exosomes (size < 200 nm), derived from the internal budding of endocytic membranes, and microvesicles (MVs) (size > 200 nm), derived from the extroflession of activated cells’ membranes [109]. Exosomes express CD63, CD81, and CD9 on their surfaces [109] and can interact with target cells through exosomal proteins and cellular receptors, the direct fusion of the exosomal membrane with the cell membrane, or endocytosis [110]. 

Ultracentrifugation is the gold standard technique for exosomes’ isolation and extraction, alone or in combination with other methods, such as density gradient centrifugation for purification. Other physical methods include polymer precipitation, size exclusion chromatography, and ultrafiltration. Otherwise, exosomes can be detected by antibodies directed against specific markers exposed on their membranes, or by the combination of physical and biological methods for EV detection, such as immunoaffinity chromatography [111].

Through autocrine and paracrine mechanisms, exosomes enable the interaction between tumor cells and TME, with immune-modulatory effects and the stimulation of the epithelial to mesenchymal transition that favors vascular invasion and metastatization [112]. Moreover, increasing evidence suggests that the interaction between tumor-derived exosomes, tumor cells, and TME may play a significant role in the development of drug resistance to TKIs in patients with HCC [113,114,115,116,117,118]. The exposure of PD-L1 on exosomes causes the direct inhibition of T cells’ function. Exosomes expressing PD-L1 compete with cancer cells and peritumoral cells in binding ICIs; as a result, lower levels of a drug can target tumor cells, resulting in a mechanism of resistance against therapy [116]. Similar to exosomes, MVs express specific markers on their surface derived from the type of cell that generates them and contain nucleic acids and proteins that could influence several biologic processes [117]. MVs expressing Hepatocyte Paraffin 1 (HepPar1+) have been found to be higher in patients with HCC than in controls without cancer, and their lack of reduction 3 months after liver resection was observed in patients with tumor recurrence [118]. AnnexinV+ EpCAM+ Human Asialoglycoprotein Receptor 1 (ASGPR1+) MVs were able to distinguish patients with cirrhosis and liver cancer (HCC or cholangiocarcinoma) from those with no malignancy, and this was confirmed by the drop in the concentration 7 days after curative resection [119]. 

EVs contain a variety of proteins, lipids, DNAs, messenger RNAs (mRNAs), microRNAs (miRNAs), and other non-coding RNAs, such as circular RNAs (circRNAs) and long non-coding RNAs (lncRNA) [120,121]. These genetic products and proteins are similar to those expressed in the tumor tissue; thus, EVs not only reflect a cancer’s features and its dynamic changes [122] but can also regulate various cellular processes such as proliferation, survival, migration, and the inhibition of the anti-tumor response [123]. In particular, long non-coding RNAs (lncRNA) have shown a potential role in modulating immunotherapy responses via TME re-programming, leading to the exhaustion of CD8 cells, which is a well-known marker of a poor response to ICIs [124]. lncRNAs’ detection and amplification has been obtained using real-time PCR techniques or, lately, microarray technology [125]. Several signatures based on lncRNAs have been associated with a response to ICIs [126,127,128,129,130,131,132]. Furthermore, circular RNAs (circRNAs) can regulate gene expression and interfere with transcription and peptide translation through the modulation of microRNAs (miRNAs) [133]. Several methods exist to detect circRNAs, such as northern blotting and real-time PCR; droplet digital PCR, which divides a sample into 20 thousand droplets to obtain the PCR amplification of the template in each droplet, has also been applied to increase the sensitivity for the detection of low levels of circRNAs. Other methods that are used include isothermal exponential amplification, which can copy oligonucleotides within minutes with a high amplification ability [134], and rolling cycle amplification, which uses circular probes as templates to generate long single-stranded DNA or RNA of the target sequence [135]. The genotype analysis of circRNA is based predominantly on NGS, but this technique loses sensitivity with an increasing read length. Recently, a novel algorithm has been developed to overcome this problem: using full-length circular reverse transcription, a long complementary DNA strand including many copies of the circRNA target is obtained; then, the full length circRNAs are sequenced using nanopore technology. This algorithm is also able to quantify circRNA expression and the presence of mutations [136]. In HCC tissue, Circular ubiquitin-like with PHD and ring finger domain 1 (circUHRF1) RNA is upregulated. It decreases the activity and number of NK cells in tumor tissue and has been associated with a resistance to ICIs [137]. CircMET (hsa_circ_0082002), which is highly expressed in HCC tissue and exosomes compared to non-tumor cells, promotes HCC cells’ survival by acting on the miR-30-5p/Snail/dipeptidyl peptidase 4 (DPP4)/CXCL10 axis, with the consequent inhibition of CD8 T cell functions and a resistance to anti-PD-1 therapy [138]. Hsa-crc-0003288 has been demonstrated to increase PD-L1 expression in vitro by the activation of the PI3K/AKT signaling pathway, thereby promoting tumor proliferation and the epithelial to mesenchymal transition of cancer cells [139]. Hsa_circRNA_104348 is upregulated in HCC tumor cells and promotes tumor progression and invasion via the Wnt/beta catenin and miR-187–3p/RTKN2 pathways, leading to a poor response to ICIs [140]. CircRHBDD1 promotes a hypoxic environment by the upregulation of glycolysis; its expression has been analyzed in 18 patients with advanced HCC who received anti-PD-1 therapy: after 4 cycles of treatment, non-responders expressed significantly higher levels of circRHBDD1 compared to responders. To confirm the tumorigenic role of circRHBDD1, its inhibition has been studied in a xenograft model: mice with silenced circRHBDD1 cells presented a better response to anti-PD-1 therapy and an increased infiltration of CD8 cells in tumor tissue [141]. Finally, miRNAs located in EVs can derive from any type of cell, including HCC tumor cells and TME; they contribute to chronic hepatic inflammation and tumorigenesis by promoting tumor growth and immune tolerance and by influencing angiogenesis and extrahepatic spread [142]. Initially, miRNAs were isolated from cells using a phenol–chloroform extraction followed by RNA precipitation. Since this method causes a loss of RNA sequences full of GC base pairs, adding a column-based RNA absorption method reduces the presence of contaminants and stabilizes the miRNA sequences. The mirVana and miRNEasy kits are two of the available kits [143,144]. The detection of circulating miRNAs is based on quantitative PCR. Lately, the application of digital PCR, which detects the absolute level of miRNA expression with a high specificity and sensitivity even in fluids, has been proposed [145].

MiRNAs can target the 3′-UTR regions of PD-L1 mRNA; otherwise, they can act indirectly on PD-1, as observed in other non-HCC solid tumors [146,147,148]. Hsa-miR-329-3p inhibits lysine-specific dymethylase 1A (KDM1A), which increases the methylation of Myocyte Enhancer Factor 2D (MEF2D), reducing the expression of PD-L1 and blocking tumor growth, as demonstrated in a xenograft model of HCC [149]. Other miRNAs, such as miR-675-5p, can upregulate or, as in the case of miR-145, miR-194-5p, and MiR-200, downregulate PD-L1 expression in HCC TME [126,134,150,151]. MiR-34a can target the 3′-UTR regions of PD-L1, thereby reducing its ability to bind to PD-1, increasing the infiltration of CD8 T cells in tumor tissue, and activating DCs [152]. MiR-155 upregulates TIM-3, resulting in an enhanced degree of T cell exhaustion [153]. Conversely, the interaction between miR-155 and the lncRNA Nuclear-Enriched Abundant Transcript 1 (NEAT1) can interfere with tumor progression in mice, thus enhancing CD8 T cell cytolysis. Moreover, miR-449c-5p, which is expressed by NK cells, can bind to TIM-3 mRNA causing its degradation and boosting the immune response in HCC TME [132]. Despite the promising role of EVs as possible non-invasive biomarkers, no gold standard for exosome isolation exists; therefore, a remarkable difference in the terms of sensitivity among the different platforms has been reported. The costs of RNA extraction, sequencing, and characterization are other concerns [154]. 

### 4.4. Antidrug Antibodies

Following the administration of ICIs, patients may present a hyperstimulation of the immune response that leads to the production of antidrug antibodies (ADA). According to the Food and Drug Administration’s indications, ADA are usually tested in screening assays, followed by a characterization and titer quantification [155]. Among them, neutralizing antibodies (NAb) bind to the ICIs, forming an immunocomplex that is cleared by the circulation. This process may lead to a reduced efficacy of checkpoints’ blockade and to a higher sensitivity to immune adverse reactions (IRAEs) [156]. The appearance of ADA has been reported following anti-PD-1, anti-PD-L1, and anti-CTLA-4 therapy, with an incidence that differs between the ICIs classes: among anti-PD1 inhibitors, Nivolumab induced ADA expression in 11.2% of patients [157,158] compared to only 2.1% for Pembrolizumab [159]. The anti-CTLA-4 agent Ipilimumab induced ADA upregulation in 5.4% of patients [160], while the anti-PD-L1 Durvalumab induced such an upregulation in 2.9% [161] of patients; the higher levels of ADA (13–54% of patients, with NAb in 4–28% of the cases) have been reported in patients treated with Atezolizumab plus Bevacizumab at different doses and regardless of tumor histology [162]. Atezolizumab serum levels can be influenced by ADA exposure, with an average reduction of 22% in treatment activity in the ADA+ group compared with the ADA- group in vitro [163]. Atezolizumab’s efficacy is also reduced by its clearance promoted by ADA. ADA production was shown to be independent of tumor type, line of therapy, treatment dose, or administration as monotherapy or in combination with other drugs, while a male sex, Caucasian ethnicity, extended tumor burden, impaired liver function, a high level of serum C-reactive protein, NLR, and lactate dehydrogenase demonstrated a strong correlation with the development of ADA following ICIs therapy [163]. Despite these results, a meta-analysis that enrolled 7736 patients across 11 clinical trials showed no differences in terms of OS and PFS among ADA+ and ADA- patients or in ADA NAb+ versus ADA NAb-patients [164]. Another meta-analysis including 1086 patients treated with Nivolumab confirmed the absence of negative effects caused by ADA in terms of adverse reactions or a loss of efficacy [165]. According to these results, the assay of ADA as a biomarker affecting the response to immunotherapy appears to be controversial and without a demonstrated usefulness. 

The main findings of the studies concerning the novel non-invasive biomarkers of the response to immunotherapy in HCC are reported in Table 2. 

### 4.5. Gut Microbiome

The liver and the gut interact in a bidirectional way. Via the portal circulation, there is a continuous passage of microorganisms, microbially derived proteins, and metabolic products, creating a functional interplay, known as the gut–liver axis. The gut microbiome plays a crucial role in maintaining the homeostasis of this system through the regulation of metabolic pathways, the immune system function, and intestinal barrier integrity [166,167]. Under normal conditions, the interaction between pathogen-associated molecular patterns (PAMPs) expressed by the gut microbiota and Toll-like receptors (TLRs) on the host cell’s surface induces a tolerogenic response, which is important for maintaining the proper functioning of the immune system and protecting the host from potentially harmful external aggressions [168,169]. Gut microbiome-derived metabolites, such as short-chain fatty acids (SCFAs) and bile acids, are also involved in the maintenance of this balance, favoring an immune-tolerogenic environment [170,171,172,173,174,175]. In patients with cirrhosis, qualitative and quantitative changes in the gut microbiota compositions have been described, with a relative decrease in beneficial bacteria and an increase in pathological ones [176,177]. These changes are associated with an inflammatory shift in metabolic and immune processes. Indeed, the intestinal barrier’s impairment and pathological bacterial translocation, which are hallmarks of chronic liver diseases, trigger a pro-inflammatory cascade that culminates in liver-focalized and systemic inflammation [178,179,180,181,182]. With the progression of liver cirrhosis, this chronic injury with persistent inflammation results in immune exhaustion, which favors HCC development [183,184,185]. 

In recent years, the scientific interest in the characterization of the gut microbiota has rapidly grown, followed by the development of two main techniques. The first to be adopted was the sequencing of the 16s rRNA gene [186], a highly conserved bacterial gene coding for the 30S ribosome, which is used for the identification of bacterial taxa down to the genus level [187]. Metagenomic sequencing, including the more accurate shotgun sequencing, is more recent and costly, but allows for the identification of species and even strains, as well as functional information [188].

Considering its profound impact on the immune system, several findings demonstrate that the gut microbiota influences the response to immunotherapy. Commensal bacteria are fundamental in orchestrating antitumor responses in TME, and distinct bacterial species modulate different immune responses [189]. In a study conducted on patients affected by advanced non-small cell lung cancer (NSCLC) and renal cell carcinoma receiving PD-1 blockade therapies, Routy et al. demonstrated an increased abundance of *A. muciniphila* in the stool specimens of responder patients versus the non-responders. Then, fecal microbiota transplantation (FMT) from the responders into germ-free (GF) or antibiotic-treated mice was performed. After tumor induction, the mice receiving an FMT from responders had a better response to the PD-1 blockade and showed an increased expression of CXCR3+ CD4+ T cells in TME. Intriguingly, an oral *A. muciniphila* supplementation in mice receiving an FMT from the non-responder patients was able to restore the efficacy of anti-PD-1 therapy [190]. In another study conducted on patients affected by metastatic melanoma receiving anti-PD-1 immunotherapy, a stool analysis of the responder patients showed a higher alpha diversity and increased abundance of *Ruminococcaceae*. Among *Ruminococcaceae*, an increased *Faecalibacterium* abundance was associated with a prolonged PFS and an improvement in CD4 and CD8 T cells, whereas *Bacteroidales* abundance correlated with a worse outcome and an immunosuppressive pathway. Likewise, an FMT from the responder patients into the GF mice was associated with a better response to therapy and greater anti-tumor activity [191]. These findings have been reinforced by the results of another phase 1 trial (NCT03353402), in which stool samples from two donors with a previously documented response to anti-PD-1 monotherapy were transplanted into a population of 10 patients affected by metastatic melanoma refractory to anti-PD-1 agents. Three patients obtained a complete or partial response, reverting the initial drug resistance, and showed a post-treatment increase of antigen-presenting cells infiltration in the gut, suggesting that the anti-tumor immune response may start in the intestine [192]. With respect to HCC patients, Zheng et al. [193] analyzed the gut microbial composition of eight patients affected by BCLC stage C HCC receiving Camrelizumab as a second-line treatment after Sorafenib; stool samples were collected before and after 3 to 12 weeks from the beginning of treatment. The baseline gut microbiota were mainly enriched with *Bacteroidetes*, followed by *Firmicutes* and *Proteobacteria*. After an ICI treatment, the patients who showed an objective tumor response presented an overgrowth of *Proteobacteria*, with a peak after 12 weeks. Among *Proteobacteria*, *Klebsiella pneumoniae* was the main species enriched in the responders, while in the non-responders an overabundance of *Escherichia coli* was reported. Furthermore, the responders presented an increased abundance of several probiotic bacteria such as *Lactobacilli, Bifidobacterium dentium*, and *Streptococcus thermophilus*, which are known to positively influence host immunity. In addition, an increase in *Ruminococcaceae* and *A. muciniphila*, which are involved in maintaining intestinal barrier integrity, was reported among responders [193]. Another small study collected stool samples from eight patients with HCC who received the anti-PD-1 agent Nivolumab [194]. The analysis of the gut microbiota demonstrated a higher concentration of *Clostridia*, *Prevotella*, and *Ruminococcaceae* in the responders, while *Ruminococcus gnavus* was predominant in the non-responder group. *Citrobacter freundii, Azospirillum* spp., and *Enterococcus durans* were correlated with a good prognosis in terms of OS and PFS, while *Escherichia coli, Lactobacillus reuteri, Streptococcus mutans,* and *Enterococcus faecium* predicted a negative outcome. The composition of the gut microbiota in HCC patients was analyzed at the phylum level, reporting an imbalance in the *Firmicutes*/*Bacteroidetes* ratio (below 0.5 or upper than 1.5) that occurred more prevalently in the non-responders than in the responders, while a higher mean ratio of *Prevotella* spp. to *Bacteroides* spp. (P/B ratio) was clearly identified in the responders; also, the presence of *Akkermansia* was detected only in the responders. Mao J. et al. [195] analyzed the fecal samples of 65 patients affected by advanced HCC or biliary tract cancer receiving anti-PD-1 therapies. The results showed that the patients with a clinical benefit response (CBR), considered as a partial or complete response to therapy or a stable disease for a minimum of 6 months, had a relative abundance of *Lachnospiraceae bacterium-GAM*79 and bacteria from *Ruminococcaceae* family, while the *Veillonellales* predominated among the patients without any clinical benefit (NCB). Moreover, a higher abundance of *Lachnospiraceae bacterium-GAM*79 was associated with a longer PFS and OS, while bacteria from the *Veillonellaceae* family were associated with a worse clinical outcome. A dynamic analysis of the gut microbiota composition also showed a decrease in bacterial diversity among the NCB group. The importance of *Ruminococcaceae* in predicting ICIs’ efficacy was confirmed by a retrospective Chinese study, in which the enrichment of *Clostridiales/Ruminococcaceae* was reported in the responders to anti-PD-1 therapy [196]. The study also showed a positive association between a high abundance of *Faecalibacterium*, belonging to the *Ruminococcaceae* family, and a longer PFS, while an increased abundance of *Bacteroidales* was associated with a worse prognosis. 

A concern with respect to these results is that the majority of the studies included Asian patients, some of whom presented chronic hepatitis. However, a recent study including 11 Caucasian cirrhotic patients with HCC treated with Tremelimumab and/or Durvalumab demonstrated that those who achieved DCshowed a lower fecal calprotectin concentration and PD-L1 serum levels at baseline; also, the pre-treatment increased the abundance of *Akkermansia* observed in patients who achieved DC, in parallel with a reduction in *Staphylococcus, Neisseria,* and *Enterobacteriaceae* [197]. Dynamic analyses of the microbiota composition during treatment showed an inverse relationship between alpha diversity; *Akkermansia* to *Enterobacteriaceae* (AE) ratio, which was considered as a marker of dysbiosis; and calprotectin levels, reinforcing the hypothesis that intestinal inflammation plays a role in influencing clinical outcomes.

Metabolites derived from the microbiome can also contribute to modulating the response to ICIs, and can be used as biomarkers. A prospective study conducted on 52 patients with advanced solid tumors receiving Nivolumab or Pembrolizumab showed a higher fecal and plasma concentration of SCFAs among responders, and fecal propionic acid was identified as a marker of PFS [198]. A possible explanation for the immune-modulating activity of SCFAs is the inhibition of histone deacetylases (HDACs), which has been associated with a higher expression of PD-1 ligands and sustained PD-1 blockade in melanoma models [199,200,201,202]. Conversely, in patients affected by metastatic melanoma, high levels of butyrate and propionate seem to reduce the efficacy of anti-CTLA-4 therapy, with negative effects on DC maturation in mice, a reduced production of IL-2, and lower numbers of memory and ICOS+ T cells [202]. In patients with Nonalcoholic fatty liver disease (NAFLD)-related HCC, the overabundance of SCFA-producing bacteria was linked to an immunosuppressive condition, with a higher expression of T regs and a reduced cytotoxic CD8+ T cells response [203]. Accordingly, a recent study by Pfister et al. has demonstrated that, in preclinical models of nonalcoholic steatohepatitis (NASH)-induced HCC, the administration of anti-PD-1 agents induced the expansion of intratumoral CD8+ PD1+ T cells, but this phenomenon did not cause tumor regression, suggesting an impairment in the cytotoxic activity of these lymphocytes [204]. Moreover, the administration of an anti-PD-1 agent induced the development of NASH-HCC and led to an overexpression of exhausted T cells. These findings were followed by a meta-analysis of three major studies on the effect of immunotherapy on patients with non-viral HCC, which confirmed a poor prognosis in terms of the OS and PFS in these patients [11,204,205,206]. These results suggest an impaired and aberrant T cell activation in NASH patients that limits ICIs’ application, and that can be explained by a dysfunctional gut–liver axis [203,204]. 

In light of these findings, the gut microbiome’s modulation by antibiotics is a key factor to consider, because in various cancers it has been associated with a worse response to immunotherapy [207,208,209]. In particular, a recent study reported a worse survival in patients who received a prior antibiotic treatment, but not in those who had been undergoing a current antibiotic treatment, during ICI therapy [210]. However, in a study on murine models of HCC, a vancomycin administration was associated with a reduction of primary to secondary bile acid conversion, due to the depletion of Gram-positive bacteria in the gut. This study showed a positive correlation between the primary bile acid concentration and CXCR6+ NKT cells’ accumulation in the liver favoring tumor inhibition, whereas secondary bile acids had opposite effects [211]. Table 3 briefly summarizes the main findings of the studies on the role of the gut–liver axis in the response to immunotherapy.

Currently, a limitation on the use of the gut microbiota as a biomarker of ICI response is the heterogeneity of the results obtained so far [193,194,195,196,197], making it impossible to identify a reliable signature or metabolic feature. Nevertheless, further studies based on human FMT are needed to confirm its efficacy as a co-adjuvant treatment for a successful immunotherapy. The influence of diet, ethnicity, lifestyle, and chronic therapies on the relationship between gut microbiota and the host [212,213,214] are other data that need to be taken into account to further stratify patients and refine gut microbiota’s prognostic significance and therapeutic usefulness. 

The novel circulating biomarkers and the gut microbiome species whose changes are under evaluation as predictors of the response to immunotherapy in HCC patients are reported in Figure 1.

## 5. Conclusions and Future Perspectives

HCC often occurs in the setting of chronic inflammation and immune exhaustion [185,189]. Recently, immunotherapies have become part of the first-line treatment for advanced HCC [4], but several immune and phenotypical tumor features negatively influence a durable anti-tumor response. Nowadays, liver biopsy is not mandatory for the diagnosis of HCC in patients with liver cirrhosis and is mainly performed for the enrollment of patients in clinical trials; the liquid biopsy analysis of circulating tumor biproducts has emerged as a simple and reproducible tool for monitoring cancer progression and assessing pharmacological efficacy, with the possibility of multiple re-evaluations during treatments [23,117]. Studies aimed at identifying noninvasive biomarkers in easy-to-analyze body fluids such as saliva are being continuously published, but interesting results aside, the generalizability is limited by the small cohorts of patients [214,215]. Unraveling the characteristics of HCC and its response to ICIs through the analysis of samples derived from body fluids is fascinating. With the improvement of gene sequencing, the discovery of sensitive biomarkers has become widespread. In particular, ctDNA and CTCs have important prognostic implications, as they could identify mutational signatures that reflect the genomic landscape of the primary tumor [103,104,105,106]. In addition, exosomes, being able to modulate cellular communication directly and indirectly through the release of their cargos, provide important information on tumors’ mutational burden, invasiveness, and resistance during therapy to a degree surpassed by no other previous biomarker [112,113,114,115,116]. Compared with traditional AFP, it has been shown that ctDNA [81,82,83] might be a better prognostic marker of responses in patients with unresectable HCC, but these findings have not been confirmed in other studies. This could be due to the low levels of CTC and ctDNA that can be found in body fluids, which reduce their application in standard diagnostic and prognostic procedures, and to the lack of antibodies against membrane markers of these products [93,106,154]. High costs and the need for multiple platforms and technologies for the comprehensive analysis of tumor products should also be mentioned as limitations for the dissemination and application of noninvasive biomarkers [93,106,154]. The main limitation of liquid biopsy is related to the lack of standardized protocols and the limited data available. Further clinical studies are needed to define the most useful biomarkers obtained from different biological materials and to standardize the assay and characterization methods in order to provide reliable information driving therapeutic decisions [93,106,154]. 

Although the search for reliable non-invasive biomarkers of ICIs’ efficacy is a compelling clinical need, histologic evaluations still hold valuable and unique information regarding TMBs, oncogenic mutations, and TMEs, which closely influence responsiveness to ICIs [216]. The direct analysis of tumor tissue also suffers from pitfalls, as biopsy specimens do not always encode the full set of tumor characteristics; constructing multimodal predictive scores that include genomic, transcriptomic, proteomic, and immunophenotypic features of the tumor, taking into account the host’s characteristics and environmental modifiers, would probably be the most promising approach for identifying the optimal responders to immunotherapy. Growing evidence suggests that the use of prognostic scores based on the combination of multiple noninvasive biomarkers, or the association between invasive and noninvasive biomarkers, and the integration of new NGS and AI technologies in clinical research could overcome these concerns [55,72,73,217,218]. The application of bioinformatics to medical research has opened the field of hepato-oncology to innovative perspectives. Artificial intelligence (AI) can be used to extract complex information from visual data derived from digitized histological samples, and it is becoming a key tool for predicting prognoses and responses to treatment in gastrointestinal cancers, including HCC [217,218]. In this scenario, an important role can be played by radiomics, which allows for the analysis of tumor heterogeneity and characteristics derived from medical imaging in a multidimensional way using quantitative features. Radiomics can extract data related to TME and its cellularity, such as the infiltration of CD8 T cells, intratumoral lymphocytes, and macrophages; provide information about qualitative and quantitative immune checkpoints’ expression; and predict responses to immunotherapy [219,220,221,222]. Finally, the interplay between the liver and gut, as well as the influence of the gut microbiome on ICIs’ efficacy, are matters of fact in several tumor types, including HCC [191,194,195,198]. Microbiome profiling should be considered as the next frontier for an integrated evaluation of the candidates towards ICI treatment to be included in AI algorithms, whenever possible, as another piece in the puzzle. What biological specimen is the most informative and easy to use (i.e., saliva, stool, and blood) is yet to be defined, as well as the optimal timepoints for its harvesting. (After antibiotic treatment, at fixed intervals, after progression?) However, little is known about the correlation between the gut microbiome and its metabolic patterns and TME. The markers of gut barrier integrity could be another interesting field to explore in the near future; in fact, bacterial translocation is a hallmark of chronic liver disease and parallels immune dysfunction in its advanced stages, linking persistent gut-derived inflammation with the promotion of hepatocarcinogenesis [166,185,223].

In conclusion, the identification of sensitive and accurate prognostic biomarkers for the evaluation of responses to immunotherapy is a compelling and developing field in hepato-oncology. The combinatory analysis of tumor tissues’ intrinsic features, the peritumoral microenvironment, and the immunological and microbiological characteristics of the host is crucial for the development of a prognostic score capable of differentiating responders from non-responders. This dynamic characterization process better adapts to the continuous changes of tumor biology, and can prognosticate the responsiveness of HCC to ICIs in a personalized manner, tailored to the individual patient.

## Figures and Tables

**Figure 1 cancers-14-04631-f001:**
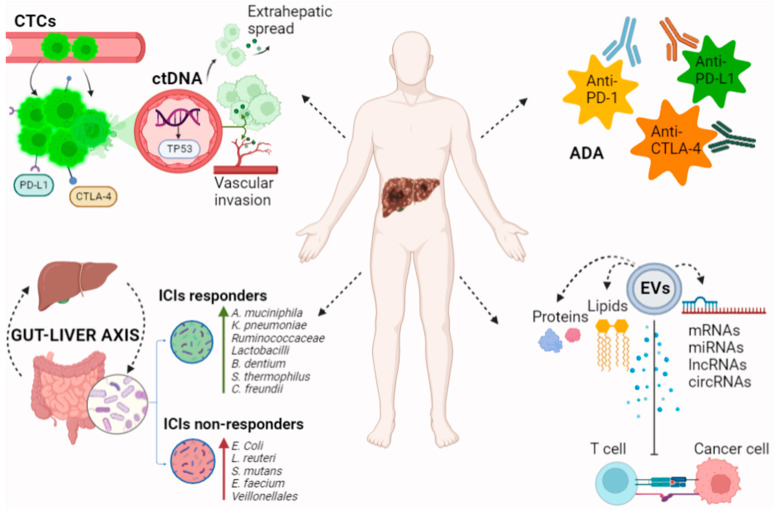
Novel biomarkers for immunotherapy in patients with HCC. CtDNA and CTCs reflect tumor growth and invasiveness of HCC. Genome analysis of ctDNA allows for the detection of prognostic tumor mutations. CTC levels correlate with tumor extension and may have a role in predicting tumor recurrence after immunotherapy. EVs contain proteins, lipids, DNAs, mRNAs, miRNAs, and non-coding RNAs including circRNAs and lncRNAs. These products influence ICIs’ efficacy by down- or up-regulating PD-L1 expression in the tumor microenvironment. Immune system hyperstimulation by ICIs can lead to the production of ADA. ADA directed against ICIs promote immunocomplex formation and drug clearance, potentially affecting anti-tumor efficacy. The gut microbiota’s composition and products regulate several immune and metabolic pathways in the gut–liver axis, including the response to ICIs. Gut microbiota profiles differ among ICIs responders and non-responders, opening the field to studies testing microbiota-targeted therapies as a new strategy in immuno-oncology. In particular increased abundance of *Akkermansia muciniphila, Klebsiella pneumoniae, Ruminococcaceae Lactobacilli, Bifidobacterium dentium, Streptococcus thermophilus,* and *Citrobacter freundii* has been linked to response to immunotherapy; *Firmicutes/Bacteroidetes* ratio between 0.5 and 1.5 and a higher mean ratio of *Prevotella* spp. to *Bacteroides* spp. are also markers of improved survival, whereas increased abundance of *Escherichia coli*, *Lactobacillus reuteri, Streptococcus mutans, Enterococcus faecium,* and *Veillonellales,* and a reduction in bacterial *diversity* has been associated with non-response. ctDNA: circulating tumor DNA, CTCs: circulating tumor cells, EVs: extracellular vesicles, mRNAs: messenger RNAs, miRNAs: microRNAs, circRNAs: circular RNAs, lncRNAs: long non-coding RNAs, ICIs: immune checkpoint inhibitors, ADA: antidrug antibodies, and PD-L1: programmed death-ligand 1.

**Table 1 cancers-14-04631-t001:** Traditional non-invasive biomarkers for immunotherapy in patients with hepatocellular carcinoma (HCC).

Study	Biomarkers	Study Design	Patients	Treatment	Outcomes	Results
Myojin Y. et al. [26]	IL-6, IFN-alpha	Prospective	64, HCC	Atezolizumab plus Bevacizumab as first or second line therapy	PFS, OS	-Higher IL-6 and IFN-alpha levels associated with poor PFS and OS -Higher IL-6 levels were more frequently related to female sex; higher levels of AFP, AST, and DCP; and poor liver function
Feun L.G. et al. [30]	TGF-beta, IFN-gamma, IL-10	Phase II prospective	28, HCC	Pembrolizumab for 60–90 days at dosage of 200 mg intravenously every 3 weeks	-Primary: DCR Maintained for at least 8 weeks -Secondary: PFS, OS, ORR, duration of response, toxicity profile	-Higher TGF-beta serum levels in non-responders -TGF-beta serum levels > 200 pg/mL associated with poor response -IFN-gamma and IL-10 correlated with PD1/PD-L1 serum levels
Mocan T. et al. [37]	sPD-L1	Prospective	121, HCC	Anti-PD-1/anti-PD-L1 drugs	-Association between sPD-L1 levels, OS, and DFS	-The best cut-off value of sPD-L1 for both DFS and OS was 96 pg/mL -Patients with high sPD-L1 (>96 pg/mL) had shorter DFS and OS (HR 5.42, 95% CI 2.28–12.91, *p* < 0.001, and HR 9.67, 95% CI 4.33–21.59, *p* < 0.001) -High sPD-L1 level independently associated with mortality
Hong J.Y. et al. [43]	CD8+ cells	Prospective	60, HCC	Pembrolizumab or Nivolumab	ORR, PFS, OS	-Partial response or stable disease associated with immunological shift (increase in cytotoxic CD8+ T cells) -Elevation of CD8+ T cells after 4 weeks correlated with response
Macek Jilkova Z. et al. [44]	CD4+ PD-L1+ cells and T regulatory cells	Prospective	32, HCC	Tremelimumab	ORR	-Baseline CD4+ PD-L1+ cells positively correlated with response to anti-CTLA4 therapy -Higher level of Tregs correlated with poor outcome
Hung Y.-P. et al. [47]	PMBC	Prospective	16, HCC	Nivolumab	Immune cells changes after immunotherapy	-Percentage of total αβ T cells or CD4 T cells did not significantly change after treatment with nivolumab, and was not related to outcomes -CD8 T cells significantly increased after 4 weeks (*p* = 0.016)
Tada T. et al. [49]	NLR	Metanalysis	249, HCC	Atezolizumab plus Bevacizumab	PFS	-Baseline level of NLR (cut-off 3.21 pg/mL) independent predictor of PFS
Muhammed A. et al. [52]	NLR, PLR and PNI	Retrospective	362, HCC	Nivolumab 60.2% Pembrolizumab 45 12.4% Ipilimumab 0.3% Ipilimumab/Nivolumab 3.6% Atezolizumab 3% Durvalumab 2.2% PD-1, CTLA-4 combination 3.9% PD-1, TKI combination 6.6%	PFS, OS	-NLR ≥ 5 and PLR ≥ 300 negatively correlated with prognosis and survival -NLR ≥ 5 and PLR ≥ 300 independent prognostic factors for OS (HR 1.73, 95% CI 1.23–2.42, *p* = 0.002 and HR 1.60, 95% CI 1.6–2.40, *p* = 0.020, respectively) -PLR only independent predictor of PFS
Mei J. et al. [53]	PNI	Prospective	442, HCC	Nivolumab 6.6%, Pembrolizumab 7.7%, Toripalimab 62.7%, Sintilimab 21.3% Camrelizumab 6.3%	PLR, NLR, CRP, CAR, PNI	-PNI score prognostic indicator for OS
Yongjiang Li. et al. [55]	HCC-GRIm-Score	Retrospective	261, HCC (161 internal cohort; 80 validation cohort)	ICIs	PFS, OS	-HCC GRIm-score from 0 to 2 points correlated with better OS
Shao Y.-Y. et al. [59]	AFP	Retrospective	60 Patients of several studies with advanced HCC receiving ICIs	ICIs	PFS, OS	-Reduction in AFP correlated with better OS -Early AFP response independent predictor for OS (HR = 0.089, 95% [CI] = 0.018–0.441; *p* = 0.003 and PFS HR = 0.128, 95% CI = 0.041–0.399; *p* < 0.001)
Hsu W.-F. et al. [60]	AFP	Retrospective	95 HCC	ICIs alone or in combination with TKIs	ORR, PFS, OS	-AFP decline > 15% in the serum within the initial 3 months of ICI therapy predictor of disease control -AFP independent predictor of OS and PFS
Teng W. et al. [61]	AFP	retrospective	90, HCC	Nivolumab	ORR, PSF, OS	-Patients divided into four classes: class I rapid AFP decrease of ≥ 50% of baseline at week 4; class II AFP changes within ± 50% of baseline at week 4 that later decreased to ≥ 10% of baseline at week 12; class III AFP changes within ± 50% of baseline at week 4 without decreasing to ≥ 10% of baseline at week 12; class IV rapid AFP increase of ≥ 50% of baseline at week 4 -ORR was 47.4%, 36.0%, 7.7%, and 5.0% in class I–IV patients, respectively -Class I and class II had better ORR, PFS, and OS -AFP independent predictor for OS and PFS
Sun X. et al. [74]	AAP (AFP and PIVKA-II combined score)	Retrospective	235 HCC	ICIs	ORR, PFS, OS	-Reduction (>50% from baseline levels at 6 weeks of treatment) in AFP and PIVKA-II correlated with ORR, OS, and PFS -AAP score ≥ 2 points associated with better PFS and OS
Hatanaka T. et al. [72]	CRAFITY (AFP and CRP)	Retrospective cohort	297 HCC	Atezolizumab Bevacizumab	Radiological response, OS	-Lower scores (0–1 points) associated with better response and OS
Teng W. et al. [73]	CAR (CRAFITY plus AFP decline after 6 weeks of ICIs)	Retrospective	89 HCC	Atezolizumab 1200 mg and Bevacizumab 5–7.5 mg/kg intravenously every 3 weeks	ORR, PFS, OS	-Lower CRAFITY score and higher AFP decline associated with better survival
Cao W. et al. [77]	AFP and s-ICAM	Prospective	87, HCC	ICIs	PFS, OS	-AFP ≤ 20 μg/L or sICAM-1 ≤ 1000 μg/L before surgery or recovered to normal after surgery associated with reduced tumor recurrence rate and better OS -Synchronously elevated levels of AFP and s-ICAM-1 showed the lowest PFS and OS

IL—Interleukin, IFN—Interferon, HCC—Hepatocellular carcinoma, PFS—Progression Free Survival, OS—Overall Survival, AFP—Alpha Fetoprotein, AST—Aspartate Aminotransferase, DCP—Des-gamma-carboxy prothrombin, TGF-beta—Transforming Growth Factor beta, DCR—disease control rate, ORR—objective response rate, PD-1—programmed death 1, PD-L1—programmed death ligand 1, sPD-L1—soluble Programmed Death Ligand 1, HR—Hazard Ratio, *p*—*p* value, ICIs—immune checkpoint inhibitors, NLR—Neutrophil to Lymphocyte ratio, DFS—disease free survival, CI—confidence interval, PBMC—Peripheral Blood Mononucleate Cells, CTLA4—cytotoxic T-lymphocyte-associated protein 4, Tregs—T regulatory cells, Platelet to lymphocytes ratio, PNI—prognostic nutritional index, LDH—Lactate Dehydrogenase, AST to ALT ratio—aspartate aminotransferase to alanine aminotransferase ratio, HCC-GRIm-Score—Gustave Roussy Immune Score, TKIs—tyrosine kinase inhibitors, AAP AFP-ALBI-PIVKA-II score, PIVKA-II—protein induced by vitamin K absence or antagonist-II CRAFITY CRP and AFP in Immunotherapy score, CRP—C reactive protein, CAR—CRAFITY score and AFP-Response, and sICAM—soluble intercellular adhesion molecule 1.

**Table 2 cancers-14-04631-t002:** Novel non-invasive biomarkers for immunotherapy in patients with hepatocellular carcinoma (HCC).

Study	Biomarker	Study Design	Patients	Treatment	Methods	Endpoints	Results
von Felden J. et al. [83]	ctDNA	prospective	121	None	Evaluation of mutations in ctDNA	-Primary endpoint: PFS stratified by mutation profiles in ctDNA. -Secondary endpoints: OS and ORR	-*TERT* promoter (51%), TP53 (32%), CTNNB1 (17%), PTEN (8%), AXIN1, ARID2, KMT2D, and TSC2 (each 6%) were the most frequent mutations in ctDNA. -Mutations in PI3K/MTOR pathway is associated with reduced PFS after TKIs but not after ICIs. -WNT mutation had no impact on survival
Oversoe S.K. et al. [84]	ctDNA	prospective	95 HCC, 45 liver cirrhosis without HCC	None	Evaluation of mutation of *TERT* in serum and tissue samples of HCC patients compared to serum of nonHCC patients	Correlation between TERT mutation and prognosis	-Plasma *TERT* C228T mutation was identified in 44% of HCC patients but in none of the non-HCC patients -*TERT* mutation was detected in 68% of liver biopsies -*TERT* mutation was associated with increased mortality when detected in plasma (adjusted HR 2.16 (1.20–3.88), *p* = 0.010) but not in tumor tissue. -*TERT* mutation in plasma correlates with higher TNM and vascular invasion
Kim S.S. et al. [85]	ctDNA	prospective	59	None	Sequencing and detection of single-nucleotide variants in ctDNA associated with prognosis	OS	-Four SNVs were frequent in ctDNA: MLH1 (13%), STK11 (13%), PTEN (9%), and CTNNB1 (4%), -Three candidate SNVs were detected in 35.5% of the patients, (MLH1 chr3:37025749T > A, STK11 chr19:1223126C > G, and PTEN chr10:87864461C > G.) -MLH1 SNV, in combination with an increased ctDNA level, predicted poor overall survival and can predict prognosis in HCC patients
Shen T. et al. [86]	ctDNA	Prospective parallel cohorts	895 HCC patients divided into 3 cohorts: cohort 1, 260 patients with liver biopsy treated with hepatectomy; cohort 2, 275 patients treated with hepatectomy; cohort 3, 360 patients without hepatectomy	Liver surgery in cohorts 1 and 2	Evaluation of mutation in TP53 in ctDNA and tumor biopsy	TP53 mutations and correlation with PFS and OS	-In Cohort 1, R249S was the most frequent mutation and was associated with a worse phenotype -R249S, but not other missense mutations, was significantly associated with worse OS (*p* = 0.006) and PFS (*p* = 0.01) of HCC patients in every cohort.
Araujo D.V. et al. [91]	bTMB	phase I prospective	85	anti-PD1	Evaluation of bTMB in ctDNA and in tumor tissue	-Correlation between bTMB and TMB -Correlation between bTMB and OS	-78.9% of patients had detectable mutations in ctDNA, median range bTMB was 5 (1–53) mutations per megabase (mut/Mb). -Among the 16 patients with detectable mutations in both biopsies and ctDNA, a statistically significant correlation between bTMB and tTMB was observed (*ρ* = 0.71; *p* = 0.002). High TMB level was not associated with better survival.
Zhu G.-Q. [92]	bTMB	prospective phase I	41	Post-operative recurrence	Whole-exome sequencing was used to detect the DNA of HCC	ctDNA prediction early post-operative tumor recurrence	-47 gene mutations were identified in the ctDNA of the 41 patients analyzed before surgery. ctDNA was detected in 63.4% and 46% of the patient plasma pre- and post-surgery, respectively. -Preoperative ctDNA positivity rate was significantly lower in the non-recurrence -Median follow-up of 17.7 months; nine patients (22%) experienced tumor recurrence. -Multivariate analyses showed that the median variant allele frequency of baseline ctDNA is a strong independent predictor of RFS in individuals with HCC.
Chen J. et al. [98]	CTCs	retrospective phase I	195	None	Evaluation of CTC count and EMT classification using the CanPatrol^®^ platform	-Detection of CTCs -Evaluation of epithelial to mesenchymal transition markers and correlation with tumor characteristics of invasiveness	-CTCs were detected in 95% of the 195 HCC -Total CTCs numbers were correlated with BCLC stages, metastasis, and serum AFP levels. -The proportion of CTCs demonstrating epithelial to mesenchymal transition was associated with ages, BCLC stages, metastasis, and AFP levels.
Yu J.-J. et al. [101]	CTCs	prospective	139	Liver surgery	Collection of samples for CTCs’ analysis one day before and three days after resection	Evaluation of CTC levels before and after surgery as indicator of early recurrence after surgery	-Increase in CTC levels after surgery correlated with vascular invasion -Changes from preoperative CTCs < 2 to postoperative CTCs ≥ 2 were associated to poor OS -Patients with persistent CTC levels of ≥2 had the worst prognosis.
Xingping Ye et al. [102]	CTCs	Prospective	42	Liver surgery	CTCs were counted 1 day prior to and 30 days after surgical excision of HCC using the CanPatrol™ system.	OS PFS	-Numbers of CTCs (>2 CTCs and >5 CTCs per 5 mL peripheral blood) were associated with the Edmondson stage in HBV-related HCC prior to surgery (*p* = 0.004 and 0.014, respectively) -Postoperative CTCs counts (>2 and >5) and pre/postoperative change in CTCs counts were significantly associated with PFS (*p* = 0.02, 0.009, and 0.001, respectively), but not with OS -Pre/postoperative changes in the CTCs count were a better predictor of performance than absolute count.
Winograd P. et al. [103]	CTCs	prospective, case control	87 patients with HCC (49 early-stage, 22 locally advanced, and 16 metastatic), 7 patients with cirrhosis, 8 healthy controls	10 patients treated with anti-PD-1 therapy	CTC count and phenotypization was obtained with an antibody-based platform	Correlation between number of CTCs, expression of PD-L1 and prognosis	-PD-L1 CTCs discriminated early from locally advanced/metastatic HCC -Regarding CTCs, patients with PD-L1+ CTCs had significantly inferior overall survival (OS) (median OS = 14.0 months vs. not reached, hazard ratio [HR] = 4.0, *p* = 0.001) -PD-L1+ CTCs resulted in an independent predictor of OS (HR = 3.22, *p* = 0.010) In patients with HCC receiving anti-PD-1 therapy, there was positive association with the presence of PD-L1+ CTCs and response.
Yue C. et al. [104]	CTCs	Prospective	35 patients with different advanced gastrointestinal tumors	Anti-PD-1 therapy	Immunofluorescence assay for semi-quantitative assessment of the PD-L1 expression levels on CTCs with four categories (PD-L1 negative, PD-L1 low, PD-L1 medium and PD-L1-high)	Correlation between levels of expression of PD-L1 on CTCs and propensity to positively response to immunotherapy (DCR)	-PD-L1-high patients had higher DCR levels -Count changes of total CTCs, PD-L1 positive CTCs, and PD-L1-high CTCs correlate with disease outcome (*p* < 0.001, *p* = 0.002 and 0.007, respectively). -PD-L1-high CTC levels at baseline correlate with progression free survival (PFS)
Winograd P. et al. [105]	CTCs	prospective case control	92 patients (8 healthy controls, 11 chronic liver disease without HCC, 73 patients with HCC).	A subgroup treated with immunotherapy	Detection of total number of CTCs and evaluation of expression of several markers, such as PD-L1 positivity	Determination of total CTCs and detection of PD-L1 positive CTCs and their correlation with response to therapy	-PD-L1+ CTCs identified with high-specificity HCC patients with early stage and advanced/metastatic disease (sensitivity = 67.7%, specificity = 92.3%, *p* < 0.0001) -Patients with PD-L1 positive CTCs who received immunotherapy showed positive response to treatment
Abbate V. et al. [118]	Exosomes	prospective case control	15 patients with HCC 5 cirrhotic patients 10 healthy subjects	Liver resection	Evaluation of circulating HepPar1+ microparticles by flow cytometry	Prognostic significance of detection of HepPar+ microparticles after surgery	-Patients with HCC showed higher levels of HepPar1+ MPs at baseline (*p* < 0.01). -HCC patients showing higher levels of HepPar1+ MPs before liver resection was presented early recurrence compared to those with lower levels (*p* = 0.02).
Julich Haerthel H. et al. [119]	Exosomes	prospective case control	172 patients with liver cancers (HCC or cholangiocarcinoma), 54 with cirrhosis and 202 controls	None	Fluorescence activated scanning to detect microparticles positive for AnnexinV+ EpCAM+ CD147+	Accuracy of AnnexinV+ EpCAM+ ASGPR1+ CD133+ microparticles in tumor detection and its prognostic value	-AnnexinV+ EpCAM+ CD147+ microparticles were elevated in HCC and CCA -AnnexinV+ EpCAM+ ASGPR1+ CD133+ were not expressed by cirrhotic and healthy controls -AnnexinV+ EpCAM+ ASGPR1+ taMPs level decreased at 7 days after curative R0 tumour resection, suggesting close correlations with tumour presence
Ji J. et al. [124]	lnc-RNA	retrospective case control	55 patients with HCC, 40 healthy volunteers	None	Detection of lnc-RNA	Role of lnc-RNA in CD8 T cells functions	-lnc-Tim3 is upregulated and negatively correlates with IFN-γ and IL-2 production in tumor-infiltrating CD8 T cells of HCC patients. -lnc-Tim3 stimulates CD8 T exhaustion and the survival of the exhausted CD8 T cells.
Li L. et al. [126]	lnc-RNA	retrospective case control	371 HCC 50 controls	None	Evaluation of lnc-RNA expression in HCC tissues compared to controls.	OS, tumor response	-lncRNA signatures resulted an independent prognostic factor for OS -lncRNAs could predict the clinical response to immunotherapy.
Xu Q. et al. [127]	lnc-RNA	retrospective randomized case control	370	None	Identification of lncRNAs signatures	Identification of lncRNAs signatures that could predict survival	-Seven immune-related lncRNA signatures were validated and resulted in independent predictive biomolecular factors -NRAV was significantly upregulated in HCC cell lines and it may serve as a key regulator in HCC
Zhou P. et al. [128]	lnc-RNA	retrospective	RNA sequences of HCC patients derived from the cancer genome atlas	None	Construction of a model of immune related lnc-RNA markers of tumor microenvironment, response to immune checkpoint blockers	Patient risk stratification and impact on survival according to lnc RNA expression in HCC patients	-Six immune-related lncRNAs were validated -NRAV showed the ability to stratify patients into high-risk and low-risk groups with significantly different survival rates -The immune-related six-lncRNA signature was a novel independent prognostic factor in HCC patients.
Zhang Y. et al. [130]	lncRNA	prospective	Training set of 368 patients and external validation cohort of 115 patients with HCC	None	Construction of lncRNA immune-related signatures via Cox regression analysis	Correlation between lncRNA immune-related signatures and response to immunotherapy, disease progression, and survival	-Expression of lnc-RNA immune-related signatures stratify patients into high or low risk of disease progression and worse survival -lnc-RNA signatures resulted in independent prognostic biomarkers -They could identify patients eligible for immunotherapy
Huang X.-Y. et al. [138]	circ RNA	retrospective case control	Human HCC cell line from 209 HCC patients and matched non tumor cells	None	Amplification of 43 circRNA in 7q21–7q31 region	Identification of circRNAs that mediate development of HCC	-circMET (hsa_circ_0082002) was overexpressed in HCC tumors and induces its proliferation and induces an epithelial to mesenchymal transition -circMET influences microenvironment through the miR-30-5p/Snail/ dipeptidyl peptidase 4(DPP4)/CXCL10 axis -Combination of the DPP4 inhibitor sitagliptin and anti-PD1 antibody improved antitumor immunity in immunocompetent mice
Xu G. et al. [139]	circRNA	retrospective	Human HCC cell line, 40 HCC tissue	None	hsa_circ_0003288 expression measured by qRT-PCR.	Regulation and function of hsa_circ_0003288 on PD-L1 during EMT and HCC invasiveness.	-hsa_circ_0003288 promoted EMT and invasion of HCC via the hsa_circ_0003288/miR-145/PD-L1 axis through the PI3K/Akt pathway -Overexpression of hsa_circ_0003288 increased PD-L1 levels and promoted EMT, migration, and invasiveness
Huang G. et al. [140]	circRNA	prospective	Human HCC cell line from 60 HCC tissue	None	HCC cell line and HCC tissue, circ RNA measurement via qRT-PCR	Regulation and functions of Has_circ_104348 and its influence on HCC development	-Has_circ_104348 was highly expressed in HCC tissue and cells, promoting proliferation and invasion of HCC -miR-187-3p negatively influences Has_circ_104348 expression
Cai J. et al. [141]	circRNA	prospective parallel cohorts	cohort I 96 HCC patients cohort II 160 HCC patients	None	HCC cell line and HCC tissue, circ RNA measurement via qRT-PCR	Regulation and function in HCC development	-circRHBDD restricts anti-PD-1 therapy in HCC -circRHBDD1 is highly expressed in anti-PD1 responder HCC patients, and targeting circRHBDD1 improves anti-PD-1 therapy in an immune-competent mouse model
Wang Y. et al. [149]	miRNA	retrospective	HCC cell line	None	qRT-PCR detection of miRNA	Influence of miR-329-3p on PD-L1 expression in HCC	miR-329-3p inhibits tumor cellular immunosuppression and reinforces the response of tumor cells to T cell-induced cytotoxic effect by targeting KDM1A mRNA
Liu Z. et al. [150]	miRNA	retrospective	152	None	qRT-PCR detection of PD-L1	Impact of EGFR-signaling PD-L1 in HCC cells	EGFR-P38 MAPK axis could up-regulate PD-L1 through miR-675-5p
Yan K. et al. [153]	miRNA	prospective case control	20 patients with HCC 20 healthy controls	None	Serum samples for PMBC analysis, qRT-PCR detection of NEAT and Tim-3	Interaction among NEAT1 and miR-155 in Tim-3 modulation in HCC patients	-NEAT1 and Tim-3 were up-regulated in the PBMCs of patients with HCC compared with healthy subjects -Down-regulation of NEAT1 enhances the cytolysis activity of CD8 T cells, miR-155 upregulates Tim-3
Wu B. et al. [163]	ADA	meta-analysis	4500 patients from 12 clinical trials across different tumor types, treatment settings, and dosing regimens	Immunotherapy with Atezolizumab/Bevacizumab	ADA screening assay before first drug administration and for other 9 cycles before the drug injection	-Risk factors for development of ADA -Role of ADA in immunotherapy efficacy	-Male sex, Caucasian ethnicity, extended tumor burden, impaired liver function, high serum CRP, NLR, and LDH had a strong correlation with the development of ADA -ADA may influence tumor response to immunotherapy but data are still controversial.

ctDNA—circulating tumor DNA, HCC—Hepatocellular Carcinoma, PFS—Progression Free Survival, TERT—Telomerase Reverse Transcriptase, TP53—Tumor Protein 53, CTNNB1—CTNN Beta Cathenin 1, PTEN—Phosphatase and TENsin homolog, KMT2D—Histone-lysine N-methyltransferase 2D, TSC2—Tuberous Sclerosis Complex 2, PI3K/mTOR—Phosphatidylinositol 3-kinase (PI3K)/Mammalian target of Rapamycin (mTOR), TKIs—Tyrosine Kinase Inhibitors, ICIs—Immune Checkpoint Inhibitors, HR—Hazard Ratio, *p*—*p* value, TNM—tumor node metastases, SNV—single nucleotide variants, MLH1—MutL protein homolog 1, STK11—Serine/threonine kinase 11, OS—Overall Survival, PD-1—Programmed Death 1, bTMN—blood Tumor Mutational Burden, TMB—Tumor Mutational Burden, mut/MB—mutations per megabase, RFS—recurrence-free CTC-Circulating Tumor Cells, EMT—Epithelial to Mesenchymal Transition, BCLC—Barcelona Clinic Liver Cancer, AFP—Alpha Fetoprotein, HBV—Hepatitis B Virus, PD-L1—Programmed Death Ligand 1, DCR—Disease Control Rate, HepPar1+—Hepatocyte Paraffin 1, MPs—Microparticles, EpCAM—Epithelial cell adhesion molecule, CD—cluster differentiation, ASGAR1—Asialoglycoprotein receptor 1, lncRNA—long non-coding RNA, TIM-3—T-cell immunoglobulin and mucin-domain-containing-3, IFN—Interferon, IL—Interleukin, NRAV—Negative Regulator of Antiviral Response, circRNA—Circular RNA, DPP4—dypeptidil peptidase 4, CXCL—The chemokine (C-X-C motif) ligand, miRNA—microRNA, qRT-PCR—quantitative real-time Polymerase Chain Reaction, KDM1A—Lysine-specific histone demethylase 1A, mRNA—messenger RNA, EGFR—Epithelial Growth Factor Receptor, MAPK—Mitogen-activated protein Kinase, PMBC—Peripheral Blood Mononucleate Cells, NEAT—Nuclear Paraspeckle Assembly Transcript 1,ADA—Antidrug Antibodies, CRP—C-reactive Protein, NLR—Neutrophil to Lymphocyte ratio, and LDH—Lactate Dehydrogenase.

**Table 3 cancers-14-04631-t003:** Studies evaluating the gut microbiota as biomarkers in patients treated with immune checkpoints inhibitors (ICIs).

Study	Patients	Treatment	Methods/Endpoints	Results
Zheng Y. et al. [193]	8 Asian patients	Camrelizumab as second-line treatment after Sorafenib	Analysis of gut microbiota and correlation with ORR	-Patients with ORR presented an overgrowth of *Proteobacteria*, with a peak after 12 weeks -Among *Proteobacteria, Klebsiella pneumoniae* was the main species enriched in responders, while in non-responders an overabundance of *Escherichia coli* was reported -Responders presented an increased abundance of *Lactobacilli, Bifidobacterium dentium*, *Streptococcus thermophilus,* and *A. muciniphila*
Chung M.-W. et al. [194]	8 Asian patients	Nivolumab	Analysis of gut microbiota and correlation with OS and PFS	-*Citrobacter freundii, Azospirillum* spp., and *Enterococcus durans* correlated with longer OS and PFS -*Escherichia coli, Lactobacillus reuteri, Streptococcus mutans,* and *Enterococcus faecium* predicted a negative outcome -lmbalance in the *Firmicutes*/*Bacteroidetes* ratio (below 0.5 or upper than 1.5) occurred more frequently in non-responders than in responders -Higher mean ratio of *Prevotella* spp. to *Bacteroides* spp. (P/B ratio) identified responders
Mao J. et al. [195]	65 Asian patients	anti-PD-1 therapies	Analysis of gut microbiota and correlation with OS and PFS	-Patients with a partial or complete response for almost 6 months presented higher levels of *Lachnospiraceae bacterium-GAM*79 and bacteria from the *Ruminococcaceae* family and these data correlate with PFS and OS -*Veillonellales* were higher in non-responders and were associated with a worse clinical outcome -Reduction in bacterial diversity among non-responders
Ponziani F.R. et al. [197]	11 caucasian cirrhotic patients	Tremelimumab and/or Durvalumab	Analysis of fecal calprotectin concentration, markers of intestinal permeability and bacterial translocation, and PD-L1 serum at baseline and following therapy and correlation with response Microbiota composition at baseline and following therapy and correlation with response	-Lower fecal calprotectin and serum PD-L1 at baseline associated with response -Higher levels of *A. muciniphila* at baseline were related to response -Dynamic changes in gut microbiota, markers of dysbiosis and intestinal permeability during treatment
Nomura M. et al. [198]	52 patients with several solid tumors	Nivolumab or Pembrolizumab	Evaluation of SCFAs levels in fecal and serum samples PFS	-Higher levels of SCFA in feces and serum samples were associated with longer PFS -Fecal acetic acid (hazard ratio [HR], 0.29; 95% CI, 0.15–0.54), propionic acid (HR, 0.08; 95% CI, 0.03–0.20), butyric acid (HR, 0.31; 95% CI, 0.16–0.60), valeric acid (HR, 0.53; 95% CI, 0.29–0.98), and plasma isovaleric acid (HR, 0.38; 95% CI, 0.14–0.99) positively correlate with PFS
Behary J. et al. [203]	90 patients: 32 with NAFLD-HCC, 28 with NAFLD-cirrhosis and 30 non-NAFLD control	All subjects with NAFLD-HCC underwent surgical resection	Evaluating compositional and functional modification of the gut microbiome occurring with the development of HCC	-Patients with NAFLD-HCC and NAFLD-cirrhosis had reduced α-diversity -*Enterobacteriaceae* numbers are higher in NAFLD-HCC compared to NAFLD-cirrhosis (*p* = 0.033) and non-NAFLD controls (*p* = 0.025) -*Bacteroides caecimuris* (*p* < 0.0001) and *Veillonella parvula* (*p* = 0.002) numbers were both significantly enriched in NAFLD-HCC, compared to NAFLD cirrhosis and non-NAFLD controls -Bacterial genes involved in SCFA synthesis from dietary fibre characterized the microbiome of NAFLD-HCC -NAFLD-HCC, but not NAFLD-cirrhosis, microbiota caused the expansion of effector IL-10+ Tregs, and reduced the expansion of cytotoxic CD8+ T cells

BCLC—Barcelona clinic liver cancer, HCC—hepatocellular carcinoma, ORR—objective response rate, OS—overall survival, PFS—progression-free survival, P/B ratio—*Prevotella* spp. to *Bacteroides* spp. ratio, PD-L1—programmed death-ligand 1, PD-1—programmed death-1, A/E—*Akkermansia* to *Enterobacteriaceae* ratio, SCFA—short chain fatty acids, NAFLD—nonalcoholic fatty liver disease, HR—Hazard Ratio, CI—Confidence Interval, IL—Interleukin, and Tregs—T regulatory cells.

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
