# Peer review of "Non-Invasive Biomarkers for Immunotherapy in Patients with Hepatocellular Carcinoma: Current Knowledge and Future Perspectives"

_cancers, 2022, doi:10.3390/cancers14194631_

Round 1
Reviewer 1 Report
The manuscript is interesting and discussed potential biomarkers that might be associated with immunotherapy for HCC. However, there are several issues: (1) Page 4, line 169, missing numbering? (2) Page 4, lines 175-177, and Page 6, table 1, is the reference 41 correct? (3) There is a lot of mistyping in this manuscript.
Author Response
1) Page 4, line 169, missing numbering?
The sentence identified by your comment (page 4, line 169) was originally conceived as the title of the following paragraph. We amended this mistake creating section 3.2 and adding the sentence as section title.
2) Page 4, lines 175-177, and Page 6, table 1, is the reference 41 correct?
The paper by “Ogihara et al.” cited in page 4, lines 175-177 of the previous version of our paper was correctely reported as ref 41; currently You can find it in page 4, lines 186-188 as ref 48 due to the insertion of new references in the revised paper. Moreover, as noted by the Reviewer, we have modified the reference in page 6, table 1 that was uncorrect in the original version; after the revision You’ll find in the table the right reference concerning the paper by Hung et al. (ref 47 of the revised version of our manuscript).
3) There is a lot of mistyping in this manuscript.
We apologize for the typing errors. We have revised the text and correted typos.
Reviewer 2 Report
The manuscript contains an extensive review of the non-invasive biomarkers for immunotherapy in HCC and the authors have significantly discussed all the current markers along with some future prospects.
However, as a researcher, I would like to see more information on the drawbacks and challenges faced with the existing biomarkers that urge for the discovery of new biomarkers.
Also a part of discussion on the different methods used for non-invasive biomarker discovery would be useful from a researcher/ readership point of view.
Author Response
1) However, as a researcher, I would like to see more information on the drawbacks and challenges faced with the existing biomarkers that urge for the discovery of new biomarkers
We thank the Reviewer for his/her general opinion on the manuscript, which has been implemented adding in the text and in the discussion considerations about the limitations of the current biomarkers (page 3 lines 99-101, page 3 lines 126-129, page 5 lines 218-221, page 5 lines 235-242, page9, lines 340-343, page 10, lines 393-397, page 12 lines 500-504, page 23, lines 698-705, page 26, lines 747-766).
2) Also a part of discussion on the different methods used for non-invasive biomarker discovery would be useful from a researcher/ readership point of view.
Thanks to the Reviewer for providing this suggestion. We have added in the text the main methods for detection, isolation and characterization of novel biomarkers (page 3, lines 94-95, page 3 lines 137-139, page 5 lines 237-239, page 9 lines 296-302, page 9 lines 326-329, page 10, lines 351-364, page 11 lines 407-413, page 11 lines 440-441, page 11 lines 445-452, page 12 lines 453-457, page 12 line 478-486, page 13 lines 507-509, page 20 lines 582-585, page 21 lines 586-587); we believe that this is useful for the reader to understand their applicability not only for research purposes but also in the clinical setting.
Reviewer 3 Report
Non-invasive biomarkers for immunotherapy in patients with hepatocellular carcinoma: current knowledge and future perspectives
Maria Pallozzi , Natalia Di Tommaso , Valeria Maccauro , Francesco Santopaolo , Antonio Gasbarrini , Francesca Romana Ponziani * , Maurizio Pompili *
Cancer Biomarkers
Keywords: biomarkers; gut microbiota; hepatocellular carcinoma (HCC); immunotherapy; liquid biopsy; PD-1; PD-L1
Dear authors,
Entitle “Non-invasive biomarkers for immunotherapy in patients with hepatocellular carcinoma: current knowledge and future perspectives” manuscript was very interesting to read current HCC biomarkers. It proves useful information non-invasive biomarkers and its related treatments. If the authors revise the manuscript minorly as I recommended, it would be more valuable to read it.
<<Minor>>
1. Line 76 and 646, “noninvasive” word should be changed to “non-invasive”.
2. I would like you to add the following references “Salivary miRNAs as non-invasive biomarkers of hepatocellular carcinoma: a pilot study” PeerJ. 2022 Jan 5;10:e12715. doi: 10.7717/peerj.12715. “Serum milk fat globule-EGF factor 8 (MFG-E8) as a diagnostic and prognostic biomarker in patients with hepatocellular carcinoma”. Sci Rep. 2019 Oct 31;9(1):15788. doi: 10.1038/s41598-019-52356-6. “Salivary Metabolites are Promising Non-Invasive Biomarkers of Hepatocellular Carcinoma and Chronic Liver Disease”. Liver Cancer Int. 2021 Aug;2(2):33-44. doi: 10.1002/lci2.25. Epub 2021 May 20.
3. Figure 1 should be more informatively revised.
Thank you so much.
Author Response
1) Line 76 and 646, “noninvasive” word should be changed to “non-invasive”.
The text was amended accordingly.
2) I would like you to add the following references “Salivary miRNAs as non-invasive biomarkers of hepatocellular carcinoma: a pilot study” PeerJ. 2022 Jan 5;10:e12715. doi: 10.7717/peerj.12715. “Serum milk fat globule-EGF factor 8 (MFG-E8) as a diagnostic and prognostic biomarker in patients with hepatocellular carcinoma”. Sci Rep. 2019 Oct 31;9(1):15788. doi: 10.1038/s41598-019-52356-6. “Salivary Metabolites are Promising Non-Invasive Biomarkers of Hepatocellular Carcinoma and Chronic Liver Disease”. Liver Cancer Int. 2021 Aug;2(2):33-44. doi: 10.1002/lci2.25. Epub 2021 May 20.
As suggested, we added the suggested references in the manuscript.
3) Figure 1 should be more informatively revised.
Thanks for this suggestion. We modified Figure 1 making it more informative.
Round 2
Reviewer 1 Report
There are still several issues in the revised manuscript as below:
1. For authorship, who's address is "2"
2. Page 4, lines 184 and 185, the unit for NLR is pg/mL?
3, Page 5, lines 235-237, the AFP description is incorrect.
Author Response
- For authorship, who's address is "2"
We addressed it to the Authors
- Page 4, lines 184 and 185, the unit for NLR is pg/mL?
We amended the mistake
- Page 5, lines 235-237, the AFP description is incorrect.
We modified the description of AFP to be more precise
Reviewer 2 Report
The authors have substantially improved the manuscript and I regard it acceptable in its current format.
Author Response
Thank you for your suggestions.